# VGMOcc: Sparse Gaussian Occupancy Prediction with Visual Geometry Model Priors

## Abstract

Accurate 3D scene understanding is essential for embodied intelligence, with occupancy prediction emerging as a key task for reasoning about both objects and free space. Existing approaches largely rely on depth priors (e.g., DepthAnything) but make only limited use of 3D cues, restricting performance and generalization. Recently, Visual Geometry Models (VGMs) such as VGGT have shown strong capability in providing rich 3D priors, yet their outputs are constrained to visible surfaces and fail to capture volumetric interiors. We present VGMOcc, a framework that adapts VGM priors for monocular occupancy prediction. Our method extends surface points inward along camera rays to generate volumetric samples, which are represented as Gaussian primitives for probabilistic occupancy inference. To handle streaming input, we further design a training-free incremental update strategy that fuses per-frame Gaussians into a unified global representation. Experiments on Occ-ScanNet and EmbodiedOcc-ScanNet demonstrate significant gains: VGMOcc improves mIoU by +9.99 in the monocular setting and +11.79 in the streaming setting over prior state of the art. Under the same depth prior, it achieves +6.73 mIoU while running 2.65× faster. These results highlight that VGMOcc effectively leverages VGMs for occupancy prediction and generalizes seamlessly to alternative 3D priors. Code will be released.

## 1 Introduction

Embodied AI agents are increasingly expected to acquire accurate and detailed 3D understanding of their surroundings Liu et al. (2025a), which is fundamental for reasoning, planning, and interaction in complex environments. In this process, vision plays a central role by providing rich semantic and geometric cues, and recent progress in vision-based 3D scene understanding has been substantial Li et al. (2024a;b); Wu et al. (2022); Rukhovich et al. (2022); Peng et al. (2024). Among various scene representations, occupancy prediction Yu et al. (2024); Wu et al. (2024) has gained particular traction by providing a unified and flexible volumetric model of both foreground objects and background structures, serving as a core building block for downstream tasks such as robotic navigation Liu et al. (2024), interactive manipulation, and autonomous driving Liu et al. (2025b); Wei et al. (2024).

While vision-centric occupancy prediction has been extensively studied in autonomous driving Li et al. (2023a); Wei et al. (2023); Huang et al. (2023); Shi et al. (2024); Wang et al. (2024a); Huang et al. (2024b;a), fine-grained occupancy prediction in indoor scenarios remains considerably more challenging and less explored. The difficulty arises from cluttered spatial layouts and the wide diversity of object categories. Recent methods such as ISO Yu et al. (2024) and EmbodiedOcc Wu et al. (2024) have made progress by incorporating depth priors Yang et al. (2024a;b). For instance, ISO lifts 2D image features into dense 3D volumes using estimated depth distribution, as shown in Figure 1(a). The volumetric features are then processed with a 3D U-Net to predict the final occupancy. In contrast, EmbodiedOcc initializes Gaussian primitives randomly and refines them through iterative cross-attention with image features. By projecting 3D anchors into images and subsequently splatting the refined Gaussians into occupancy prediction, as illustrated in Figure 1(b). Although effective, these approaches make only limited use of depth priors and incur substantial redundancy by representing vast empty regions, which ultimately constrains performance and generalization.

Concurrently, Visual Geometry Models (VGMs) such as VGGT Wang et al. (2025b) have emerged as powerful 3D vision foundation models Wang et al. (2024b); Leroy et al. (2024); Yang et al.

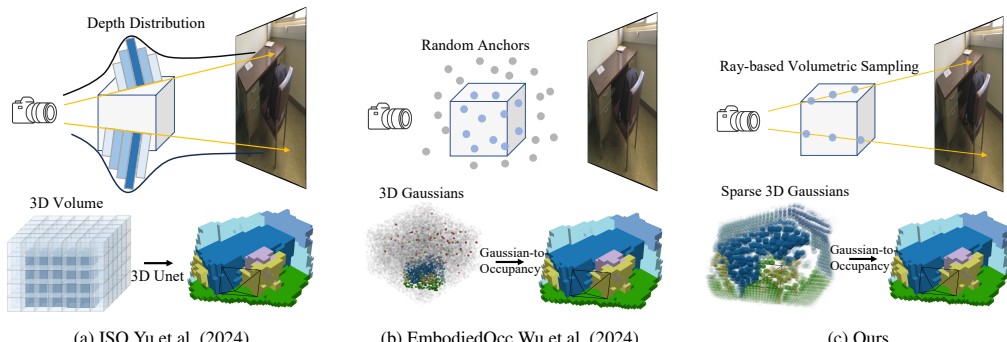

(a) ISO Yu et al. (2024)   (b) EmbodiedOcc Wu et al. (2024)   (c) Ours

Figure 1: **Comparison of monocular occupancy prediction pipelines.** ISO Yu et al. (2024) formulates depth estimation as a multi-class classification problem, using the predicted depth distributions to lift 2D image features into dense 3D volumes, which are then processed by a 3D U-Net for occupancy prediction. EmbodiedOcc Wu et al. (2024), by contrast, initializes random 3D anchors and applies cross-attention to aggregate image features, predicting Gaussian primitives that are splatted into voxels. Many of these Gaussians fall in empty regions, shown as gray primitives. In contrast, VGMOcc employs ray-based volumetric sampling to generate sparse Gaussians concentrated on or within objects, producing a compact and efficient representation for occupancy inference.

(2025); Wang et al. (2025d); Wu et al. (2025). VGMs capture rich scene attributes, including depth, point maps, and camera parameters, and enable high-quality 3D reconstruction, thereby providing strong 3D priors. These models offer abundant structural cues, making them a promising resource for advancing occupancy prediction. However, their outputs are inherently surface-centric: depth and point maps are restricted to visible surfaces, with each pixel typically corresponding to a single 3D surface point. As a result, volumetric interiors remain unrepresented, and directly deriving occupancy from VGM outputs is non-trivial, often resulting in limited accuracy.

To bridge this gap, we introduce VGMOcc, a new framework that advances occupancy prediction by leveraging VGM priors together with sparse Gaussians. Our approach builds on four key components: **(1)** To overcome the limitation that VGMs provide only surface points, we design a *ray-based volumetric sampling* module that extends points inward along their camera rays, as illustrated in Figure 1(c). Each extended point predicts a Gaussian primitive that captures its local region. **(2)** We adopt an *opacity-based pruning* strategy that discards low-opacity Gaussians, significantly reducing redundancy with negligible performance loss (Table 5). **(3)** Occupancy is inferred from the remaining sparse Gaussians using a probabilistic formulation following Huang et al. (2024a). **(4)** To adapt the framework to embodied scenarios with streaming input, we develop a training-free *incremental update strategy* that incrementally fuses per-frame Gaussians into a coherent global representation.

We evaluate VGMOcc on both the monocular Occ-ScanNet Yu et al. (2024) dataset and the streaming EmbodiedOcc-ScanNet Wu et al. (2024) benchmark. With VGGT Wang et al. (2025b) as the prior, our method surpasses the previous state of the art by 9.99 mIoU and 8.24 IoU in the monocular setting, and by 11.79 mIoU and 9.21 IoU in the streaming setting. Moreover, when using the same depth prior as EmbodiedOcc, VGMOcc achieves gains of 6.73 mIoU and 3.41 IoU, while running at $2.65\times$ FPS. These results highlight both the accuracy and efficiency of our approach.

We summarize our contributions as follows:

1. We propose VGMOcc, a novel framework for 3D occupancy prediction that combines VGM priors with sparse continuous Gaussians, enabling fine-grained volumetric prediction in challenging indoor scenarios.

2. To address the limitation that VGMs predict only visible surfaces, we introduce a ray-based volumetric sampling strategy that effectively reconstructs volumetric interiors from surface-based priors.

3. We present a sparse Gaussian to occupancy formulation with opacity-based pruning and a training-free incremental update strategy, which together improve efficiency and extend the model to streaming video inputs.

4. We conduct extensive experiments and ablations, showing that VGMOcc consistently achieves state-of-the-art performance on public datasets and generalizes effectively across different 3D priors.

## 2 RELATED WORKS

### 2.1 VISUAL GEOMETRY MODEL

Early visual geometry models such as DUSt3R Wang et al. (2024b) and MASt3R Leroy et al. (2024) predict coupled scene representations (e.g., camera poses and geometry parameterized by pointmaps) from image pairs, but require expensive post-processing and symmetric inference for unconstrained multi-view SfM. Subsequent works including Spann3R Wang & Agapito (2025), CUT3R Wang et al. (2025c), and MUSt3R Cabon et al. (2025) eliminate the reliance on classical optimization by introducing latent state memory in transformers, enabling multi-view reconstruction in a more end-to-end manner. Fast3R Yang et al. (2025) further scales this paradigm to handle over 1000 input images efficiently. Building on this line, VGGT Wang et al. (2025b) employs an attention transformer to jointly predict pointmaps, depth, poses, and tracking features, relying on minimal 3D inductive biases but leveraging large-scale training data. StreamVGGT Zhuo et al. (2025) reformulates VGGT with a causal transformer to overcome memory explosion when processing long video sequences. Several variants extend VGGT: $\pi^3$ Wang et al. (2025d) fine-tunes VGGT to remove the reliance on the first input frame as the reference coordinate system, while Dens3R Fang et al. (2025) introduces normal prediction to enrich geometric cues.

### 2.2 OCCUPANCY PREDICTION

MonoScene Cao & De Charette (2022) propelling semantic scene completion Cao & De Charette (2022); Yao et al. (2023); Song et al. (2017) into more challenging 3D occupancy with only image as input. While occupancy prediction has been extensively studied in outdoor autonomous driving scenarios Tang et al. (2024); Huang et al. (2024b;a); Li et al. (2023a), research on indoor settings remains relatively limited Yu et al. (2024); Wu et al. (2024); Wang et al. (2025a). Existing methods adopt different representations and strategies. Some approaches employ fixed 3D grids or voxel representations, where 2D image features are lifted into dense volumetric grids using depth distributions or predictions along camera rays, followed by 3D convolution or volumetric decoders Cao & De Charette (2022); Yao et al. (2023); Yu et al. (2024); Li et al. (2023b); Philion & Fidler (2020). Transformers have also been explored for volumetric representation learning Shi et al. (2024). Other methods rely explicitly on depth information or signed distance embeddings, incorporating surface locations inferred from depth into dense 3D grids Song et al. (2017), while tri-plane representations have been proposed to avoid computational burden of dense volumes Huang et al. (2023). Beyond fixed grids, point-based formulations have been investigated, where methods initialize dense 3D random points and refine them through iterative updates Huang et al. (2024b); Wu et al. (2024); Wang et al. (2024a). Sparsity has also been leveraged: some methods construct dense 3D volumes and then discard empty voxels to obtain sparse voxel representations, which are subsequently processed using sparse convolutions Tang et al. (2024) or transformers Li et al. (2023a); Lu et al. (2024).

## 3 METHODS

In this section, we first review the preliminaries of occupancy prediction and Gaussian splatting in Section 3.1. We then present our framework, which leverages visual geometry model (VGM) priors to predict Gaussians and infer occupancy. As illustrated in Figure 2, given an input image, we use a VGM to extract features and generate 3D predictions. To address the limitation that VGMs predict only surface points, which is insufficient for occupancy estimation as it requires modeling object interiors, we propose a *Ray-based Volumetric Sampling* scheme (Section 3.2). This approach extends surface points into interior points by sampling along camera rays. Each sampled point, together with its associated features, is then used to predict Gaussian attributes. Next, we apply an opacity-based pruning strategy to further reduce the number of Gaussians, and splat the remaining sparse Gaussians into occupancy using a probabilistic formulation Huang et al. (2024a) (Section 3.3). Finally, to handle embodied scenarios with streaming video inputs, we introduce a simple yet effective incremental update strategy (Section 3.4).

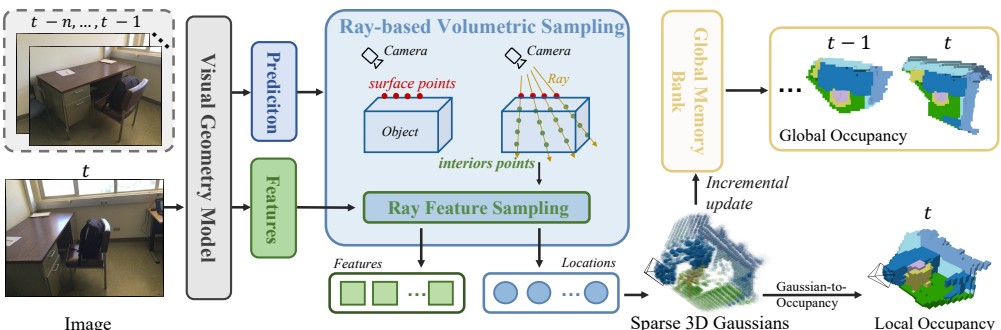

Figure 2: **Overview of VGMOcc.** Given an input RGB image, the visual geometry model (VGM) predicts surface points and extracts 3D-aware features. These surface points guide ray-based volumetric sampling to estimate interior points, which serve as Gaussian centers. The extracted features are combined with learnable embeddings to predict Gaussian attributes, and the resulting primitives are splatted to infer occupancy probabilistically. Monocular predictions are incrementally integrated into a global memory bank, enabling coherent large-scale occupancy construction.

### 3.1 PRELIMINARY

**Occupancy Prediction.** We aim to address the problem of monocular 3D occupancy prediction. Given a single RGB image $\mathbf{I}$, the goal is to predict a voxel-wise occupancy map with semantic labels $\mathbf{O} \in \mathbb{R}^{X \times Y \times Z \times N_c}$, where $X$, $Y$, and $Z$ denote the spatial resolution of the 3D scene, and $N_c$ is the number of semantic labels.

**Gaussian Splatting.** A 3D scene can be compactly represented by semantic Gaussian primitives $\mathbf{G} = \{\mathcal{G}_i\}_{i=1}^{P}$, where each primitive $\mathcal{G}_i$ describes a local region centered at its mean $\mu_i$, and is parameterized by scale $s_i$, rotation $r_i$, opacity $a_i$, and semantic feature $c_i$. This representation is both continuous and efficient, supporting differentiable rendering and compact encoding of scene geometry and semantics.

**Gaussian-to-Occupancy.** Occupancy can be inferred from Gaussian primitives via Gaussian-to-voxel splatting, which aggregates the contributions of neighboring Gaussians around a voxel $p$ Wu et al. (2024); Huang et al. (2024b;a). Formally, we have:

$$\hat{o}(p; \mathbf{G}) = \sum_{i \in \mathcal{N}(p)} g_i(p; \mu_i, s_i, r_i, a_i, c_i) \qquad (1)$$

where $\mathcal{N}(p)$ denotes the set of Gaussians that influence voxel $p$.

### 3.2 RAY-BASED VOLUMETRIC SAMPLING

Foundational Vision Geometry Models (VGMs) Wang et al. (2024b; 2025b); Yang et al. (2025); Wu et al. (2025), such as VGGT, , reconstruct 3D scenes by predicting depth and point maps that capture only the *visible surfaces*. However, occupancy prediction requires reasoning about both surfaces and *volumetric interiors*, which is critical for embodied AI tasks such as navigation and manipulation. Surface-only reasoning fails to capture the inherent thickness of real-world objects.

To address this limitation, we propose a *ray-based volumetric sampling* strategy. The core idea is to extend the predicted surface depth values along the corresponding camera rays, thereby approximating the interior volumes of objects. Formally, given a single RGB image $\mathbf{I} \in \mathbb{R}^{H \times W \times 3}$, we extract image features $\mathbf{F} \in \mathbb{R}^{H \times W \times C}$ using a pretrained VGM to exploit the learned 3D prior. Since depth is predicted at full resolution in VGMs (which is computationally costly), we instead use intermediate features before the depth prediction layer, apply downsampling, and regress depth using a lightweight MLP:

$$\mathbf{F} = \text{VGM}(\mathbf{I}), \quad \mathbf{F}^{\frac{1}{4}} = \text{DownSample}(\mathbf{F}), \quad \mathbf{d} = \text{MLP}(\mathbf{F}^{\frac{1}{4}}) \qquad (2)$$

where $\mathbf{F}^{\frac{1}{4}} \in \mathbb{R}^{\frac{H}{4} \times \frac{W}{4} \times C}$ denotes the downsampled feature map with spatial resolution reduced by a factor of 4, and $\mathbf{d} \in \mathbb{R}^{\frac{H}{4} \times \frac{W}{4}}$ is the predicted depth map. Given pixel coordinate $(u, v)$ in image

space, we can calculate the normalized camera ray direction $\mathbf{r}_{(u,v)}$ using camera intrinsics:

$$x = \frac{u - c_x}{f_x}, \quad y = \frac{v - c_y}{f_y}, \quad \mathbf{r}_{(u,v)} = \frac{[\,x,\,y,\,1\,]^\top}{\sqrt{x^2 + y^2 + 1}} \tag{3}$$

where $(c_x, c_y)$ are the principal point offsets, $(f_x, f_y)$ are the focal lengths, and $\mathbf{r}_{(u,v)}$ denotes the normalized camera ray direction at $(u, v)$. To model volumetric thickness, we sample $K$ points along the ray beyond the surface point $\mathbf{x}_{\text{surf}}$. Specifically, for pixel $(u, v)$:

$$\mathbf{x}_{(u,v)}^{\text{surf}} = \mathbf{d}_{(u,v)} \cdot \mathbf{r}_{(u,v)}, \quad \mathbf{x}_{(u,v,k)} = \left(\mathbf{d}_{(u,v)} + \delta_k\right) \mathbf{r}_{(u,v)}, \quad k = 1, \ldots, K, \tag{4}$$

where $\mathbf{d}_{(u,v)}$ is the depth at $(u, v)$, and $\delta_k$ denotes the offset for the $k$-th sample, defined as

$$\{\delta_k\}_{k=1}^K = \text{linspace}(0, 1, K) \cdot \text{scale}(\cdot) \tag{5}$$

Here, $\text{scale}(\cdot)$ is dynamically predicted to adapt to varying object sizes. To predict Gaussian attributes for each sampled point, we first extract its features from the image feature map $\mathbf{F}^{\frac{1}{4}}$. To facilitate this, we introduce a learnable embedding matrix $\mathbf{E} \in \mathbb{R}^{K \times C}$, which is randomly initialized and implemented with nn.Embedding in PyTorch Paszke et al. (2019). The point-wise features are obtained via broadcast addition:

$$\hat{\mathbf{F}}^{\frac{1}{4}} = \mathbf{F}^{\frac{1}{4}} \oplus \mathbf{E}, \quad \hat{\mathbf{F}}^{\frac{1}{4}} \in \mathbb{R}^{\frac{H}{4} \times \frac{W}{4} \times K \times C} \tag{6}$$

where $\oplus$ denotes broadcast addition and $C$ is the feature dimension. Finally, another MLP is applied to predict Gaussian attributes:

$$\{\mathcal{G}_i\} = \{s_i, r_i, a_i, c_i\} = \text{MLP}(\hat{\mathbf{F}}^{\frac{1}{4}}), \text{ where } i = 1, ..., \frac{H}{4} \times \frac{W}{4} \times K \tag{7}$$

By augmenting surface-based predictions with ray-based volumetric sampling, our framework generates a richer set of 3D points, producing more faithful occupancy representations. This design overcomes the limitations of surface-only models like VGGT and eliminates the need for dense 3D anchors Wu et al. (2024) or lifting 2D features into full 3D volumes Yu et al. (2024), as required by prior approaches.

### 3.3 FROM SPARSE GAUSSIANS TO OCCUPANCY PREDICTION

To cover the 3D space, prior work Wu et al. (2024) initializes a dense set of predefined 3D Gaussian anchors and classifies each as "occupied" or "non-occupied" to construct occupancy maps. Since most voxels in real-world scenes are empty, this strategy allocates a large fraction of primitives to non-occupied space, resulting in redundancy and inefficiency, as illustrated in Figure 3(a). In contrast, our approach leverages ray-based volumetric sampling (Section 3.2), which naturally places Gaussians on and within objects, yielding sparse yet expressive distributions, shown in Figure 3(b). To infer occupancy from these sparse Gaussians, we adopt the probabilistic Gaussian superposition formulation from GaussianFormer2 Huang et al. (2024a). In this formulation, regions without nearby Gaussians or far from any primitive, are naturally classified as empty voxels. Specifically, for a query point $p$, the contribution of a Gaussian $\mathcal{G}_i$ is:

$$o(p; \mathcal{G}_i) = \exp\left(-\tfrac{1}{2}(p - \mu_i)^\top \Sigma_i^{-1}(p - \mu_i)\right) \tag{8}$$

where $\mu_i$ is the mean of the $i$-th Gaussian and $\Sigma_i$ is

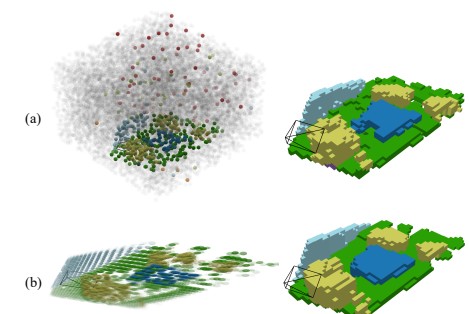

Figure 3: **Comparison of Gaussian representations.** (a) EmbodiedOcc, where gray indicates Gaussians predicted as empty. A substantial portion of primitives are placed in empty space, resulting in an inefficient representation. (b) Our method is much more compact where Gaussians are concentrated in occupied regions.

its covariance matrix, derived from the predicted rotation $r_i$ and scale $s_i$. $o(p; \mathcal{G}_i)$ denotes the occupancy probability at location $p$ induced by Gaussian $\mathcal{G}_i$, which smoothly decays to zero as $p$ moves away from $\mu_i$.

To further improve efficiency, we prune primitives with negligible opacity. Specifically, we discard all Gaussians with opacity $a_i < \tau$ (default $\tau = 0.01$), ensuring that only informative primitives contribute to the final occupancy aggregation.

## 3.4 Incremental Gaussians Update

For embodied agents that gradually perceive a scene through progressive exploration, it is important to extend monocular predictions to streaming video. To this end, we design a training-free post-processing strategy that incrementally updates Gaussians for sequential video inputs in such scenarios. We maintain a global Gaussian memory bank $\mathcal{M} = \{\mathcal{G}^t\}_{t=1}^M$, which accumulates Gaussians from frames 1 through $M$. For each incoming image frame $\mathbf{I}_t$ at timestep $t$, our monocular pipeline predicts a set of Gaussians $\mathbf{G}^t = \{\mathcal{G}_i^t\}_{i=1}^{P_t}$ where $P_t$ denotes the total number of Gaussians predicted at this timestep. To update the memory bank, for each $\mathcal{G}_j$ in the memory $\mathcal{M}$, we search for Gaussian neighbors $\mathcal{N}(\mathcal{G}_j) \subseteq \mathbf{G}^t$ within a spatial radius $\epsilon$ from the center of $\mathcal{G}_j$. If neighbors $\mathcal{N}(\mathcal{G}_j)$ are found, we fuse them by weighted averaging:

$$\theta_i \leftarrow \frac{\gamma\, p_i \theta_i + (1-\gamma) \sum_{\mathcal{G}_j \in \mathcal{N}(\mathcal{G}_i)} p_j \theta_j}{\gamma\, p_i + (1-\gamma) \sum_{\mathcal{G}_j \in \mathcal{N}(\mathcal{G}_i)} p_j} \tag{9}$$

where $\theta \in \{\mu, \Sigma, a, c\}$ denotes the mean, covariance, opacity, and semantic feature, respectively, $p$ is top-1 class confidence, and $\gamma \in (0,1)$ controls the temporal weighting between memory and new Gaussians. We set $\gamma < 0.5$ so that newer Gaussians receive higher weight. If no neighbors are found, the new Gaussians $\mathbf{G}^t$ are directly inserted into $\mathcal{M}$. This incremental strategy integrates temporal information without retraining, while naturally incorporating uncertainty-aware fusion and temporal weighting for robustness.

## 3.5 Training Losses

We optimize our model with a composite objective that balances classification, segmentation, and geometric supervision. Specifically, we combine focal loss $L_{\text{focal}}$, Lovász-Softmax loss $L_{\text{lov}}$, and scene-class affinity losses $L_{\text{scal}}^{\text{geo}}$ and $L_{\text{scal}}^{\text{sem}}$, following EmbodiedOcc Wu et al. (2024), to guide voxel-wise occupancy prediction. Unlike prior approaches Wu et al. (2024), which rely on an external pretrained depth estimator, we add a Huber loss $L_{\text{depth}}$ directly on the predicted depth to enable end-to-end optimization of the entire pipeline, which strengthens geometric consistency between depth and occupancy, and eliminates the overhead of depth pretraining.

$$\begin{aligned} \mathcal{L} = {} & L_{\text{focal}}\big(Y_{\text{mono}}^{\text{fov}}, Y_{\text{gt}}^{\text{fov}}\big) + L_{\text{lov}}\big(Y_{\text{mono}}^{\text{fov}}, Y_{\text{gt}}^{\text{fov}}\big) \\ & + L_{\text{scal}}^{\text{geo}}\big(Y_{\text{mono}}^{\text{fov}}, Y_{\text{gt}}^{\text{fov}}\big) + L_{\text{scal}}^{\text{sem}}\big(Y_{\text{mono}}^{\text{fov}}, Y_{\text{gt}}^{\text{fov}}\big) + L_{\text{depth}} \end{aligned} \tag{10}$$

## 4 Experiments

### 4.1 Datasets and Metrics

**Occ-ScanNet** Yu et al. (2024) is a large-scale benchmark for monocular indoor occupancy prediction, containing 45,755 training samples and 19,764 testing samples. The dataset covers diverse scenes and viewpoints, and provides voxelized frames in $60 \times 60 \times 36$ grids, corresponding to a $4.8\text{m} \times 4.8\text{m} \times 2.88\text{m}$ volume in front of the camera. Each voxel is annotated with 12 semantic classes, including 11 valid categories (ceiling, floor, wall, window, chair, bed, sofa, table, TV, furniture, objects) and one class for empty space.

**EmbodiedOcc-ScanNet** Wu et al. (2024) is a reorganized version of Occ-ScanNet, consisting of 537 training scenes and 137 validation scenes. Each scene contains 30 posed frames, and the global occupancy resolution of a scene is defined as $\frac{l_x \times l_y \times l_z}{(0.08\text{m})^3}$ where $l_x \times l_y \times l_z$ denotes the spatial range of the scene in world coordinates.

**Evaluation Metrics.** We follow prior work and adopt mIoU and IoU as evaluation metrics. Specifically, for **Occ-ScanNet** Yu et al. (2024); Wu et al. (2024), we compute IoU between predictions and ground truth within the camera frustum of each frame. For **EmbodiedOcc-ScanNet** Wu et al. (2024), we evaluate global occupancy by computing IoU at the scene level, where the entire reconstructed scene is considered.

## 4.2 IMPLEMENTATION DETAILS

We employ the AdamW optimizer Loshchilov & Hutter (2017) with a weight decay of 0.01. The learning rate is linearly warmed up during the first 1000 iterations to a maximum value of $2 \times 10^{-4}$ and then decayed following a cosine schedule. The model is trained for 10 epochs with a total batch size of 8 on 4 NVIDIA A800 GPUs. Input images are resized such that the longer side is 518 pixels, following the setting in VGGT Wang et al. (2025b). We apply gradient clipping with a maximum norm of 1.0. Unless otherwise specified, we adopt VGGT as the default visual geometry model, set $K = 16$ for ray-based volumetric sampling, and $\tau = 0.01$ for opacity-based pruning.

Table 1: **Monocular prediction performance on the Occ-ScanNet dataset.**

| Method | IoU | ceiling | floor | wall | window | chair | bed | sofa | table | tvs | furniture | objects | mIoU |
|---|---|---|---|---|---|---|---|---|---|---|---|---|---|
| TPVFormer Huang et al. (2023) | 33.39 | 6.96 | 32.97 | 14.41 | 9.10 | 24.01 | 41.49 | 45.44 | 28.61 | 10.66 | 35.37 | 25.31 | 24.94 |
| GaussianFormer Huang et al. (2024b) | 40.91 | 20.70 | 42.00 | 23.40 | 17.40 | 27.0 | 44.30 | 44.80 | 32.70 | 15.30 | 36.70 | 25.00 | 29.93 |
| MonoScene Cao & De Charette (2022) | 41.60 | 15.17 | 44.71 | 22.41 | 12.55 | 26.11 | 27.03 | 35.91 | 28.32 | 6.57 | 32.16 | 19.84 | 24.62 |
| ISO Yu et al. (2024) | 42.16 | 19.88 | 41.88 | 22.37 | 16.98 | 29.09 | 42.43 | 42.00 | 29.60 | 10.62 | 36.36 | 24.61 | 28.71 |
| Surroundocc Wei et al. (2023) | 42.52 | 18.90 | 49.30 | 24.80 | 18.00 | 26.80 | 42.00 | 44.10 | 32.90 | 18.60 | 36.80 | 26.90 | 30.83 |
| EmbodiedOcc Wu et al. (2024) | 53.55 | 39.60 | 50.40 | 41.40 | 31.70 | 40.90 | 55.00 | 61.40 | 44.00 | 36.10 | 53.90 | 42.20 | 45.15 |
| EmbodiedOcc++ Wang et al. (2025a) | 54.90 | 36.40 | 53.10 | 41.80 | 34.40 | 42.90 | 57.30 | 64.10 | 45.20 | 34.80 | 54.20 | 44.10 | 46.20 |
| VGMOcc(ours) | **63.14** | **51.67** | **59.93** | **52.07** | **46.44** | **51.35** | **64.45** | **69.47** | **54.30** | **51.76** | **63.29** | **53.36** | **56.19** |

Table 2: **Embodied prediction performance on the EmbodiedOcc-ScanNet dataset.**

| Method | IoU | ceiling | floor | wall | window | chair | bed | sofa | table | tvs | furniture | objects | mIoU |
|---|---|---|---|---|---|---|---|---|---|---|---|---|---|
| TPVFormer Huang et al. (2023) | 35.88 | 1.62 | 30.54 | 12.03 | 13.22 | 35.47 | 51.39 | 49.79 | 25.63 | 3.60 | 43.15 | 16.23 | 25.70 |
| SurroundOcc Wei et al. (2023) | 37.04 | 12.70 | 31.80 | 22.50 | 22.00 | 29.90 | 44.70 | 36.50 | 24.60 | 11.50 | 34.40 | 18.20 | 26.27 |
| GaussianFormer Huang et al. (2024b) | 38.02 | 17.00 | 33.60 | 21.50 | 21.70 | 29.40 | 47.80 | 37.10 | 24.30 | 15.50 | 36.20 | 16.80 | 27.36 |
| SplicingOcc Wu et al. (2024) | 49.01 | 31.60 | 38.80 | 35.50 | 36.30 | 47.10 | 54.50 | 57.20 | 34.40 | 32.50 | 51.20 | 29.10 | 40.74 |
| EmbodiedOcc Wu et al. (2024) | 51.52 | 22.70 | 44.60 | 37.40 | 38.00 | 50.10 | 56.70 | 59.70 | 35.40 | 38.40 | 52.00 | 32.90 | 42.53 |
| EmbodiedOcc++ Wang et al. (2025a) | 52.20 | 27.90 | 43.90 | 38.70 | 40.60 | 49.00 | 57.90 | 59.20 | 36.80 | 37.80 | 53.50 | 34.10 | 43.60 |
| VGMOcc(ours) | **61.41** | **42.61** | **51.35** | **51.49** | **48.72** | **54.32** | **67.91** | **70.73** | **52.94** | **54.75** | **64.76** | **49.67** | **55.39** |

## 4.3 OCCUPANCY PREDICTION RESULTS

**Results on Occ-ScanNet.** As shown in Table 1, our method achieves substantial improvements over prior approaches on Occ-ScanNet. In terms of overall performance, VGMOcc reaches 63.14 IoU and 56.19 mIoU, outperforming EmbodiedOcc by +11.04 mIoU and EmbodiedOcc++ by +9.99 mIoU. Besides, VGMOcc consistently surpasses competing methods across all indoor classes. These results highlight that our framework makes highly effective use of VGGT priors, enabling more accurate modeling of indoor structures and yielding more reliable occupancy predictions.

**Results on EmbodiedOcc-ScanNet.** Table 2 reports the embodied prediction performance on the EmbodiedOcc-ScanNet benchmark. Our method achieves a substantial improvement over existing approaches, reaching 61.41 IoU and 55.39 mIoU, which is +9.2 and +11.8 higher than the previous SoTA EmbodiedOcc++ Wang et al. (2025a). These results demonstrate that VGMOcc effectively exploits VGM priors to maintain coherent scene representations over time, making it well-suited for streaming and embodied settings.

## 4.4 ABLATION STUDIES

**Comparison with Different VGM Priors.** To further validate our approach, we instantiate it with the DepthAnything backbone following Yu et al. (2024); Wu et al. (2024). Unlike these methods, which require additional modules to form a full pipeline, **Ours-DPT** attaches only a lightweight head to the backbone. As shown in Table 3, this variant achieves better performance while revealing a favorable trade-off among accuracy, efficiency, and model size. With the same depth prior, **Ours-DPT** consistently outperforms EmbodiedOcc Wu et al. (2024), improving IoU from 53.55 to 56.96 and mIoU from 45.15 to 51.88, while running nearly $3\times$ faster (28.22 vs. 10.66 FPS) and

Table 3: **Comparison on the Occ-ScanNet dataset in terms of accuracy, efficiency, and model complexity.** DPT, $\pi^3$, and VGGT refer to Yang et al. (2024b), Wang et al. (2025d), and Wang et al. (2025b), respectively. All FPS values are measured on the same computer with NVIDIA A800 GPU, averaged over 1000 runs after an initial warm-up of 100 iterations.

| Model | IoU | mIoU | FPS | #Params | #Gaussians |
|---|---|---|---|---|---|
| ISO Yu et al. (2024) | 42.16 | 28.71 | 3.63 | 303.05M | $\times$ |
| EmbodiedOcc Wu et al. (2024) | 53.55 | 45.15 | 10.66 | 231.45M | 16201 |
| VGMOcc (Ours-DPT) | 56.96 | 51.88 | **28.22** | **97.95M** | 11128 |
| VGMOcc (Ours-VGGT) | 63.14 | 56.19 | 5.26 | 942.31M | **5876** |
| VGMOcc (Ours-$\pi^3$) | **63.25** | **56.74** | 5.93 | 827.23M | 12351 |

requiring substantially fewer parameters (97.95M vs. 231.45M) and Gaussians (11,128 vs. 16,201). We also assess stronger geometry priors. Using $\pi^3$ Wang et al. (2025d), **Ours-$\pi^3$** attains the best performance in the table (63.25 IoU / 56.74 mIoU), but at the cost of requiring substantially more Gaussians compared to VGGT. Overall, these results show that our framework is plug-and-play across 3D priors: it translates stronger priors ($\pi^3$, VGGT) into substantial accuracy gains, while the DepthAnything-based variant offers performance–accuracy balance.

Table 4: **Ablation study on the number of sampled points $K$ along each ray.** Increasing $K$ improves accuracy, but the gain saturates beyond $K = 16$ despite a large increase in number of Gaussians.

| $K$ | mIoU | IoU | #Gaussians |
|---|---|---|---|
| 1 | 47.88 | 53.10 | 3079 |
| 2 | 52.65 | 56.35 | 2310 |
| 4 | 55.28 | 60.35 | 2731 |
| 8 | 55.76 | 61.67 | 3940 |
| 16 | 56.19 | 63.14 | 5876 |
| 32 | 56.72 | 63.84 | 20206 |

Table 5: **Ablation study on the opacity threshold $\tau$.** Lower thresholds (e.g., $> 0.01$) achieve the best performance, while higher values prune too many Gaussians and degrade accuracy.

| Threshold | mIoU | IoU | #Gaussians |
|---|---|---|---|
| $> 0.01$ | 56.19 | 63.14 | 5876 |
| $> 0.02$ | 55.21 | 62.08 | 3509 |
| $> 0.03$ | 54.59 | 61.45 | 2468 |
| $> 0.04$ | 54.29 | 61.09 | 1931 |
| $> 0.05$ | 54.16 | 60.84 | 1612 |
| $> 0.06$ | 54.02 | 60.54 | 1398 |
| $> 0.07$ | 53.82 | 60.13 | 1239 |
| $> 0.08$ | 53.51 | 59.62 | 1116 |
| $> 0.09$ | 53.12 | 59.01 | 1014 |
| $> 0.10$ | 52.65 | 58.31 | 930 |

**Effect of number of sampled points $K$ along each ray.** We study the impact of sampled number $K$ in Table 4. Increasing $K$ consistently improves performance, as more samples enrich the coverage of volumetric interiors. However, the improvement saturates beyond $K = 16$, where IoU and mIoU exhibit only marginal gains at $K = 32$ despite a substantial increase in the number of Gaussians.

**Effect of opacity threshold $\tau$.** We conduct ablation studies on the choice of opacity threshold $\tau$ as shown in Table 5. A smaller threshold (e.g., $> 0.01$) yields the best performance, achieving 56.19 mIoU and 63.14 IoU. As $\tau$ increases, performance gradually degrades due to excessive pruning.

**Joint effect of $\tau$ and $K$.** We evaluate different opacity thresholds (0.01–0.10) under varying $K$ values $1, 2, 4, 8, 16, 32$ (Figure 4). The results confirm a con-

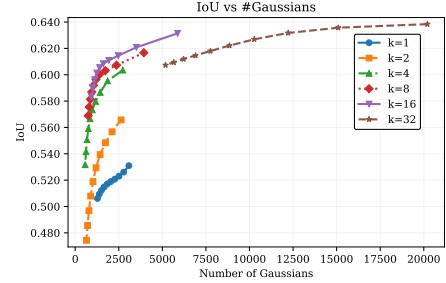

Figure 4: **IoU vs number of Gaussians.** Each curve corresponds to a fixed number of sampled points $K$. Moving from left to right along a curve reflects decreasing $\tau$ from 1.0 to 0.01.

sistent trend: lower thresholds and larger $K$ improve accuracy, but with diminishing returns. We adopt $K = 16, \tau = 0.01$ as default.

### 4.5 QUALITATIVE RESULTS

We present qualitative comparisons on monocular occupancy prediction in Figure 5, where we compare against EmbodiedOcc Wu et al. (2024). In addition, Figure 6 illustrates two examples of our incremental update strategy on streaming inputs. More visualizations are provided in the Appendix.

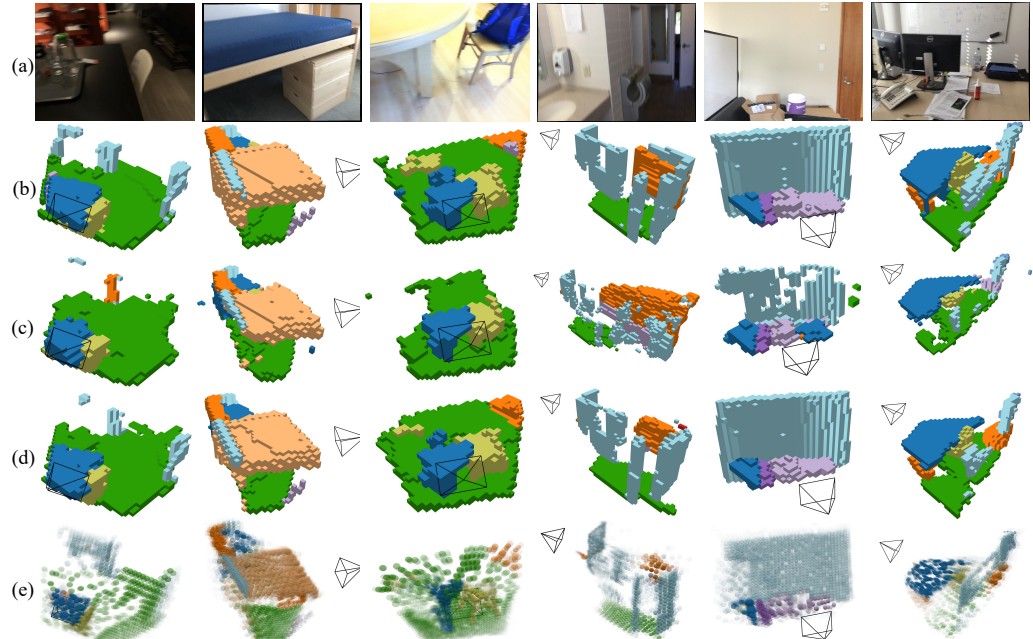

Figure 5: **Qualitative comparison on monocular occupancy prediction.** (a) shows the input RGB images, (b) the ground-truth occupancy, (c) the predictions of EmbodiedOcc Wu et al. (2024), (d) the predictions of our method, and (e) the visualization of the Gaussian primitives predicted by our method. Compared to EmbodiedOcc, our framework produces more accurate and complete reconstructions, while the Gaussian representation provides interpretable intermediate geometry.

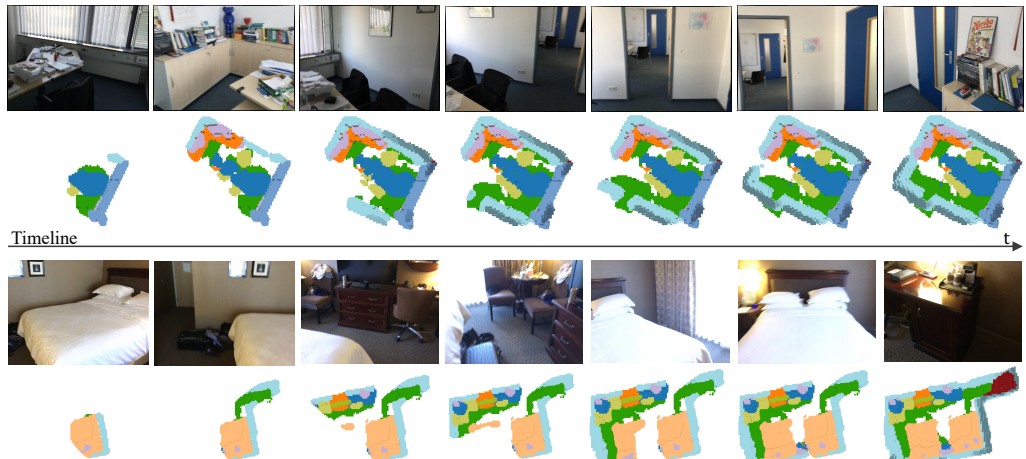

Figure 6: **Qualitative results on streaming inputs.** Our incremental update strategy progressively integrates information from sequential frames. The predictions become increasingly complete as more frames are observed, demonstrating the effectiveness of our streaming design.

## 5 CONCLUSION

We proposed VGMOcc, a novel framework that leverages visual geometry model priors for fine-grained occupancy prediction. By extending surface points along camera rays into volumetric samples, our method constructs sparse Gaussian primitives that compactly capture scene geometry. We further introduced an incremental update mechanism that adapts the framework seamlessly to streaming video inputs. Extensive experiments on Occ-ScanNet and EmbodiedOcc-ScanNet show that VGMOcc achieves SoTA accuracy and generalizes across different priors, providing a scalable and effective solution for embodied 3D perception. Looking ahead, we believe VGMOcc offers a promising step toward integrating strong geometric priors into broader embodied AI tasks, including interactive scene understanding, navigation and manipulation.

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

# A  APPENDIX

## A.1  HYPER-PARAMETERS OF INCREMENTAL UPDATE STRATEGY

By default, we use $\epsilon = 0.04$ (half of the voxel size) to aggregate Gaussian primitives and $\gamma = 0.1$ to weight Gaussian attributes. To gain deeper insight, we study the influence of these hyper-parameters in the proposed incremental update strategy, evaluating them on the EmbodiedOcc-ScanNet benchmark. The results are reported in Tables 1 to 3. Using the top-1 prediction probability during fusion yields slightly better results than without as shown in Table 1 (mIoU improves by 0.21), indicating that confidence-aware weighting stabilizes updates. For the aggregation radius $\epsilon$, a smaller value (0.02) achieves the highest mIoU and IoU, whereas a larger value (0.06) degrades performance Table 2 due to oversmoothing. Regarding the temporal weight $\gamma$, the performance remains relatively stable, likely because the dataset contains only static scenes. This observation raises an open question of how to effectively handle dynamic objects, which we leave for future research. Overall, these ablations provide a more comprehensive understanding of the proposed strategy.

Table 1: **Effect of using top-1 probability for weighting.**

|     | mIoU  | IoU   |
|-----|-------|-------|
| w/  | 55.39 | 61.41 |
| w/o | 55.18 | 61.13 |

Table 2: **Effect of the aggregation radius $\epsilon$.**

| $\epsilon$ | mIoU  | IoU   |
|------------|-------|-------|
| > 0.02     | 55.65 | 61.82 |
| > 0.04     | 55.39 | 61.41 |
| > 0.06     | 54.54 | 60.31 |

Table 3: **Effect of the temporal weight $\gamma$.**

| $\gamma$ | mIoU  | IoU   |
|----------|-------|-------|
| 0.1      | 55.39 | 61.41 |
| 0.3      | 55.36 | 61.36 |
| 0.5      | 55.37 | 61.38 |

## A.2  MORE QUALITY RESULTS

We present more visualization results in Figure 1 and Figure 2.

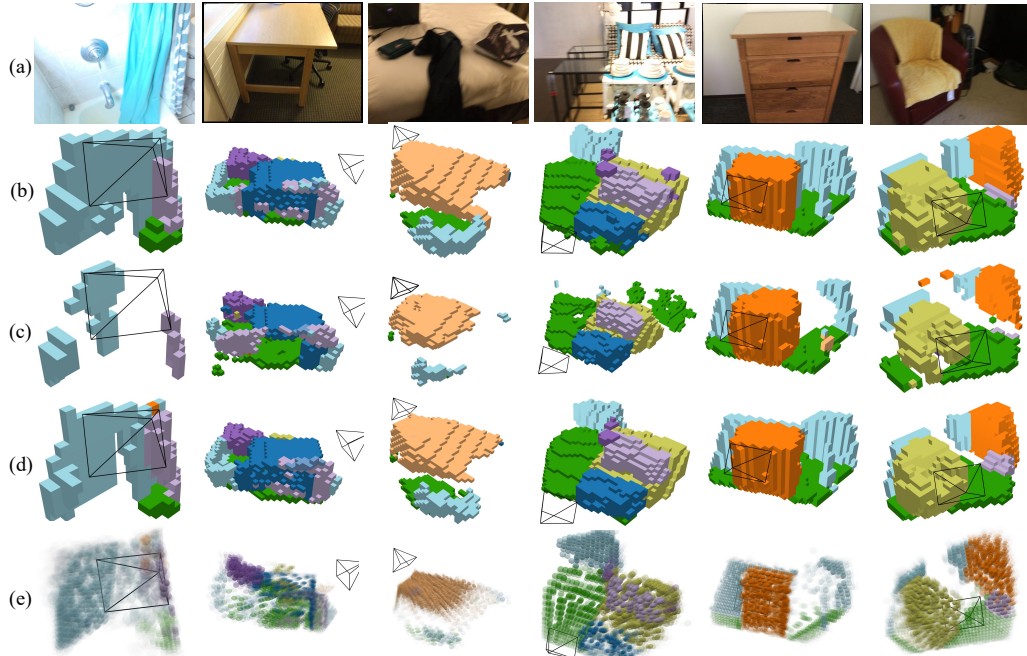

Figure 1: **Qualitative comparison on monocular occupancy prediction.** (a) shows the input RGB images, (b) the ground-truth occupancy, (c) the predictions of EmbodiedOcc Wu et al. (2024), (d) the predictions of our method, and (e) the visualization of the Gaussian primitives predicted by our method.

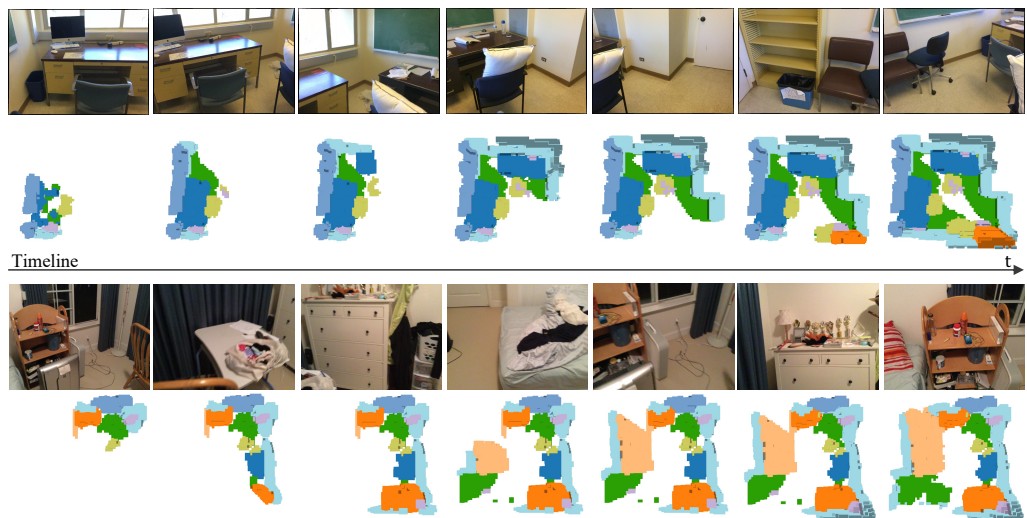

Figure 2: **Qualitative results on streaming inputs.**

## A.3 LIMITATION

While our method achieves strong performance, we note two limitations. First, the model is less accurate on large flat objects such as floors. Second, in the incremental update process, the number of Gaussian primitives keeps increasing, potentially causing inefficiency over long sequences. Both issues present promising directions for further improvement.

## A.4 LLM USAGE

We only employ LLMs to polish the writing of this paper, without using them for any other purpose.

