# OpenReview forum: "VGMOcc: Sparse Gaussian Occupancy Prediction with Visual Geometry Model Priors"
_ICLR.cc/2026/Conference — ICLR 2026 Conference Withdrawn Submission_

### Official Review · Reviewer_UuXc · 2025-10-15

**Soundness:** 3
**Presentation:** 4
**Contribution:** 2
**Rating:** 4
**Confidence:** 5

**Summary:**

This paper proposes a framework, VGMOcc, that uses VGM priors for monocular occupancy prediction.
VGMOcc initializes a certain number of Gaussians beyond the predicted low-res depths along the rays, and uses a simple network to predict the physical properties and probabilities for these Gaussians.
Experiments on Occ-ScanNet and EmbodiedOcc-ScanNet demonstrate the effectiveness of the proposed framework.

**Strengths:**

1. This paper is well-written and easy to read. The targeted problems and the related solutions are clear, and the details are kindly enough for readers who are not familiar with this field.
2. The proposed ray-based volumetric sampling strategy is intuitive and easy to understand. Results show that using this strategy can effectively reduce the number of Gaussians used.
3. The author conducted extensive experiments to support their statements.

**Weaknesses:**

1. The performance of VGMOcc largely relies on the advantages of foundational models such as VGGT. Since the foundational model is directly adopted, the performance comparison in the main table is therefore unfair (as the training costs of VGGT are very large). Moreover, this also implies that the peak GPU memory usage and the training time of VGMOcc and the baselines should also be included in the comparison. In addition, the incorporation of the foundational model VGGT clearly affects the parameter count and computational efficiency of VGMOcc, and such trade-offs should in fact be given more thorough discussion.
2. Please give more details about the method Ours-DPT in Table 3. The implementation of this method is ambiguous now.
3. If possible, please provide the experimental results of VGMOcc on NYUv2.

A small advice, details like how the learnable embedding matrix is implemented (Line 231) can be removed from the main paper, as such details are too trivial and may undermine the professionalism of the paper itself.

Please answer the above questions carefully and provide more thorough discussion and comparison. I will consider raising the score.

**Questions:**

Please refer to the Weaknesses.

**Details Of Ethics Concerns:**

No ethics concerns.

---

### Official Review · Reviewer_M93q · 2025-10-25

**Soundness:** 3
**Presentation:** 3
**Contribution:** 2
**Rating:** 4
**Confidence:** 5

**Summary:**

This paper combines visual geometry models, such as VGGT, with Gaussian-based representation for indoor 3D occupancy prediction. The paper proposes depth guided initialization, opacity pruning and streaming techniques. The VGMOcc demonstrates great improvements compared with previous SOTA methods.

**Strengths:**

1. The initialization scheme based on depth predicted from VGM prior is reasonable and effective in improving the utilization of Gaussian primitives.
2. The performance improvement is outstanding compared with previous SOTA methods.
3. The writing of the paper is clear and easy to follow.

**Weaknesses:**

1. The format for citations is wrong, i.e. the use of parentheses.
2. The novelty of this paper is limited, since the major contribution is the combination of the VGMs and EmbodiedOcc.

**Questions:**

1. The VGM model is used for depth prediction. How does it compare with depth prediction models?
2. How do you determine the thickness of objects since the VGM model only predicts depths for pixels?
3. How does the probabilistic formulation of GaussianFormer-2 affect the performance? The authors should also conduct ablation study on it.
4. EmbodiedOcc also utilizes depth information with a depth-aware branch. What is the difference between your initialization scheme and the design used in EmbodiedOcc?
5. Why is the FPS of VGMOcc lower than EmbodiedOcc?

---

### Official Review · Reviewer_qNeS · 2025-10-31

**Soundness:** 2
**Presentation:** 3
**Contribution:** 2
**Rating:** 4
**Confidence:** 5

**Summary:**

This paper introduces **VGMOcc**, a framework for monocular 3D occupancy prediction using Visual Geometry Model (VGM) priors. By extending surface points into volumetric interiors with ray-based sampling, VGMOcc generates sparse Gaussian primitives for efficient and accurate scene representation. The paper also presents incremental update strategies for streaming inputs. Experiments on Occ-ScanNet and EmbodiedOcc-ScanNet demonstrate **VGMOcc**’s effectiveness, achieving state-of-the-art performance in both monocular and streaming settings.

**Strengths:**

1. The paper is well-organized and easy to understand, with clear explanations and intuitive figures that illustrate the methodology.
2. The experiments show strong performance, demonstrating the potential of the proposed framework in indoor 3D occupancy prediction tasks.

**Weaknesses:**

1. The paper's novelty is limited. The main components of the framework, including the Visual Geometry Model (from VGGT [1] & $π^3$ [2]), Gaussian-to-Occupancy (from GaussianFormer-2 [3]), and Incremental Gaussian Update (from EmbodiedOcc [4]), are derived from prior works. The primary contribution lies in replacing random Gaussian initialization with ray-based sampling to improve representation efficiency, which makes the overall contribution incremental.
2. The experiments fail to validate the effectiveness of the paper’s main innovation. Performance improvements appear to primarily stem from using stronger visual geometry models. The lack of fair comparisons between ray-based sampling and random Gaussian initialization makes it difficult to assess the contribution of the proposed module to model performance and efficiency.
3. The experimental comparisons in Tables 1 and 2 are unfair. VGMOcc uses significantly stronger visual geometry models as backbones compared to other methods, which undermines the fairness of the reported performance improvements.

**Reference**

[1] VGGT: Visual Geometry Grounded Transformer.

[2] $π^3$: Permutation-Equivariant Visual Geometry Learning.

[3] GaussianFormer-2: Probabilistic Gaussian Superposition for Efficient 3D Occupancy Prediction.

[4] EmbodiedOcc: Embodied 3D Occupancy Prediction for Vision-based Online Scene Understanding.

**Questions:**

See weaknesses.

---

### Official Review · Reviewer_UNhj · 2025-10-31

**Soundness:** 3
**Presentation:** 3
**Contribution:** 2
**Rating:** 4
**Confidence:** 4

**Summary:**

This paper presents VGMOcc, a novel framework for predicting 3D occupancy from a single image by effectively using Visual Geometry Model (VGM) priors. Existing methods are often inefficient or limited because they only capture visible surfaces, failing to model the interior of objects. VGMOcc overcomes this limitation with its core contribution: ray-based volumetric sampling. This technique takes the surface points predicted by a VGM and extends them inward along camera rays to generate samples that represent the object's volume. These sampled points are then used to create a sparse set of Gaussian primitives, which is a much more efficient and compact representation than dense 3D grids. For handling streaming video, the framework includes a training-free incremental update strategy that fuses these per-frame Gaussians into a coherent global scene map. Experiments show that VGMOcc achieves new state-of-the-art results on the Occ-ScanNet and EmbodiedOcc-ScanNet benchmarks, significantly outperforming prior methods in both accuracy and speed.

**Strengths:**

1. This paper introduces a novel formulation for adapting surface-based priors to volumetric prediction. While other methods use depth priors, they often lift 2D features into dense 3D volumes (like ISO) or use random 3D anchors (like EmbodiedOcc), both of which are inefficient.
2. This paper presents the ray-based volumetric sampling strategy, which extends the VGM's surface predictions inward along camera rays to "fill in" the object's volume.
3. A training-free incremental update strategy that fuses per-frame Gaussians into a unified global representation is proposed.
4. The authors conduct extensive experiments and achieve impressive improvements on two standard benchmarks, Occ-ScanNet (monocular) and EmbodiedOcc-ScanNet (streaming).

**Weaknesses:**

1. The proposed incremental update strategy in Eq. 9 does not take the camera shifts among frames into account. Previous-occupancy methods, such as FBOcc, transform the former queries into the current frame using a pose-transform matrix.
2. The introduction of VGGT is actually used for Gaussian initialization. Previous methods also proposed many approaches to initialize queries/gaussians, such as OUPS (NeurIPS 2024). Please compare with these methods.
3. The VGGT features are considered to lack semantic features. The author should consider a dual geometry-semantic branch for occupancy prediction.
4. The ray-based volumetric sampling strategy relies on strong, and often incorrect, assumptions about scene geometry: (a) For objects like walls, windows, doors, or TV screens, this method will incorrectly sample $K=16$ points through the object, placing a trail of "occupied" Gaussians in what should be empty space on the other side; (b) For concave shapes (like a "U" shaped object or the inside of a bowl), a ray may hit the front surface, but sampling "inward"  will incorrectly fill the concave empty space with Gaussians; (c) The ray might hit a foreground object (e.g., a chair), but the inward sampling may place points inside a different background object (e.g., a table behind the chair).
5. The significance of the incremental update strategy is not studied, especially compared with learnable methods such as attention (transformers).
6. The framework's performance is heavily dependent on the accuracy of the initial VGM prior. The ray-based sampling is guided by the VGM's surface prediction; it does not appear to refine it.

**Questions:**

1. Is the VGGT jointly trained with the occupancy network? Please compare the results of joint training and no joint training.
2. How does the model handle thin objects like walls, windows, doors, or TV screens? The method samples $K$ points (e.g., $K=16$) inward along a ray. Does this not incorrectly place a trail of "occupied" Gaussians in the empty space or adjacent room behind these thin surfaces?
3. Would it be more robust to learn a per-pixel depth distribution (akin to ISO 5) or predict a variable number of samples $K$ based on semantic features, rather than relying on a fixed $K$ and a single scale( ) factor? This could allow the model to learn that "walls" are thin while "sofas" are thick.

---

### Note · Authors · 2025-11-13

I have read and agree with the venue's withdrawal policy on behalf of myself and my co-authors.